# Investigation of Biogenic Passivating Layers on Corroded Iron

**DOI:** 10.3390/ma13051176

**Published:** 2020-03-06

**Authors:** Lucrezia Comensoli, Monica Albini, Wafa Kooli, Julien Maillard, Tiziana Lombardo, Pilar Junier, Edith Joseph

**Affiliations:** 1Laboratory of Microbiology, Institute of Biology, University of Neuchâtel, 2000 Neuchâtel, Switzerland; lucrezia.comensoli@empa.ch (L.C.); monica.albini@cnr.it (M.A.); wafam.kooli@gmail.com (W.K.); pilar.junier@unine.ch (P.J.); 2Laboratory of Technologies for Heritage Materials, Institute of Chemistry, University of Neuchâtel, 2000 Neuchâtel, Switzerland; 3Laboratory for Environmental Biotechnology, ENAC-IIE-LBE, Ecole Polytechnique Fédérale de Lausanne, 1015 Lausanne, Switzerland; julien.maillard@epfl.ch; 4Laboratory of conservation research, Sammlungszentrum, Swiss national museum, Lindenmoosstrasse 1, 8910 Affoltern am Albis, Switzerland; Tiziana.Lombardo@nationalmuseum.ch; 5Haute Ecole Arc Conservation-Restauration, HES-SO, 2000 Neuchâtel, Switzerland

**Keywords:** iron corrosion, SEM, Raman spectroscopy, biogenic minerals, bacterial iron reduction, cultural heritage, conservation-restoration, corrosion stabilization

## Abstract

This study evaluates mechanisms of biogenic mineral formation induced by bacterial iron reduction for the stabilization of corroded iron. As an example, the *Desulfitobacterium hafniense* strain TCE1 was employed to treat corroded coupons presenting urban natural atmospheric corrosion, and spectroscopic investigations were performed on the samples’ cross-sections to evaluate the corrosion stratigraphy. The treated samples presented a protective continuous layer of iron phosphates (vivianite Fe^2+^_3_(PO_4_)_2_·8H_2_O and barbosalite Fe^2+^Fe^3+^_2_(PO_4_)_2_(OH)_2_), which covered 92% of the surface and was associated with a decrease in the thickness of the original corrosion layer. The results allow us to better understand the conversion of reactive corrosion products into stable biogenic minerals, as well as to identify important criteria for the design of a green alternative treatment for the stabilization of corroded iron.

## 1. Introduction

Cast iron and steel objects corrode rapidly when they are exposed outdoors, and complex layers composed of iron oxides and iron hydroxides are formed over time [1,2,3]. In the presence of moisture, these corrosion layers adsorb water and incorporate particulate matter and pollution agents from the atmosphere, which instigate further corrosion processes. Generally, in addition to iron oxides and iron hydroxides, iron sulfates are also frequently reported compounds within an atmospheric corrosion layer, especially in polluted and urban areas [4,5]. Near coastal areas, chlorine is a problematic element. Iron items exposed to such surroundings are susceptible to chloride-promoted corrosion, leading to chemical as well as mechanical damage to these objects [2,3,5]. Chloride ions diffused into the objects as counter-ions to iron(II) ions are produced by the oxidation of the metal [6]. Iron oxyhydroxides are then produced, including chloride-bearing akaganéite FeO_0.833_(OH)_1.167_Cl_0.167,_ and another ferrous hydroxychloride β–Fe_2_(OH)_3_Cl [6,7]. In the presence of such hygroscopic compounds, iron corrosion starts at low relative humidity (RH) values, down to 15% RH, and dramatically increases above 35% RH, leading to the fragmentation and disintegration of the objects [8]. 

Hence, outdoor iron objects that are exposed to these environments are subjected to severe degradation and would be damaged or lost without maintenance and appropriate conservation strategies. Therefore, several conservation and restoration methods are currently performed to stabilize the corrosion layers on outdoor iron-based structures. Organic coatings, such as waxes, resins, and oils, are applied to protect iron surfaces, acting as a physical barrier against atmospheric agents [3,9,10]. Nevertheless, these protective systems often need frequent maintenance to avoid failures of the coating film and exposure of the metal substrate to further oxidation [11]. Another approach employed is the use of corrosion inhibitors. However, most of these are hazardous compounds such as chromates, benzoates or nitrites, which have to be carefully manipulated [3]. Concerning archaeological iron artifacts, three methods are used to stabilize their corrosion layer. First, to remove the chloride ions from the corrosion layer, the object is immersed in anoxic and alkaline aqueous solutions [3,12]. Nevertheless, this method is based on osmotic diffusion, which is a slow process, and thus the solution needs to be replaced regularly when the chloride ions’ concentration stops increasing or stays low (below 20 ppm). Also, a large quantity of generated waste needs to be processed afterward for safe disposal. Second, electrolytic reduction allows an increase in the porosity of the corrosion layer and thus enhances the diffusion of harmful salts from the objects [13]. However, this method is restricted to large marine finds, as there is a significant loss of the surface and a lack of control over the amount of salts extracted and the corrosion products reduced during hydrogen bubbling [3]. Finally, plasma treatment is usually applied as a pre-treatment, as it creates cracks and fissures that will facilitate the diffusion of chloride ions during a successive alkaline bath [14].

For all these reasons, there is a general consensus that, to date, an efficient and durable protective system to control iron corrosion on archaeological artifacts does not exist [15].

Microbes are frequently considered detrimental to metallic surfaces, as they are associated with the process of microbial induced corrosion (MIC) [16,17,18]. However, given the large metabolic diversity found within the microbial world, an increasing number of studies focus on the exploitation of microbial processes to inhibit corrosion. Experimental evidence suggests that a biofilm of aerobic bacteria attached to copper surfaces in freshwater and seawater reduces the copper corrosion rate by decreasing the oxygen content nearby [15]. In addition, several studies reported the protective behavior of specific biofilms on submerged carbon steel pipelines [19,20,21,22]. In particular, the formation of iron phosphates as a protective layer was accomplished through the exploitation of bacterial biomineralization processes. For instance, electrochemical measurements demonstrated that the presence of a biogenic layer of vivianite (Fe^2+^_3_(PO_4_)_2_·8H_2_O) produced by *Geobacter sulfurreducens* on the surface of carbon steel coupons had a protective effect against corrosion [23].

In the examples cited above, most of the bio-based approaches were developed for the protection of bare iron surfaces before their exposure to outdoor environments. As part of our research topic, we investigated the potential of microbes for the stabilization of already corroded iron (archaeological objects and outdoor surfaces) by converting part of the reactive corrosion layer into more stable biogenic minerals (Figure 1). In particular, we studied different bacterial species, *Shewanella loihica*, *Desulfitobacterium hafniense* and *Aeromonas* spp., for their ability to produce biogenic iron minerals on corroded steel coupons [24,25,26,27,28]. Hence, vivianite and siderite were produced by *S. loihica* on costal-exposed coupons, while *D. hafniense* and *Aeromonas* spp. induced the formation of vivianite and siderite on urban-exposed coupons [26,27,28]. In order to better understand the biomineralization process involved, corroded coupons exposed in an urban environment and treated with *D. hafniense* have been investigated through the present stratigraphic study.

## 2. Materials and Methods 

### 2.1. Description of Samples

Samples of 12.5 × 25 × 2–3 mm were obtained from a steel plate with a natural urban corrosion layer mainly composed of lepidocrocite and goethite [25,26]. The plate was exposed for about 10 years in an urban environment (Zürich, Switzerland).

### 2.2. Bacterial Treatment 

The *D. hafniense* strain TCE1 (DSMZ-German Collection of Microorganisms and Cell Culture GmbH 12704) was used for this study. This bacterium was selected as it can use a variety of electron acceptors, especially halogenated organic compounds and metals [29,30]. In addition, a previous study revealed that this strain was able to reduce Fe^3+^-citrate in the presence of 0.2% and 0.3% NaCl, and that it was more efficient in terms of production of iron phosphates on the surface of corroded iron coupons [25,26]. Bacterial pre-incubation was performed in the dark at 30 °C under agitation in a standard mineral medium under anoxic conditions in 500 mL serum bottles, until reaching an optical density (OD600) of 0.1−0.15, as previously described [31]. Quantities of 45 mM of lactate and 20 mM of fumarate were added as an electron donor and acceptor, respectively, as well as a buffer solution containing phosphates and carbonates (4 mM K_2_HPO_4_ and 1 mM NaH_2_PO_4_; 54 mM NaHCO_3_ and 6 mM NH_4_HCO_3_) to maintain the pH at 7.3. To avoid the corrosion of iron coupons during treatment that would lead to a misinterpretation of the results obtained, passivating conditions were achieved by replacing O_2_ with a mix of N_2_/CO_2_ (80%/20%) and by adding Na_2_S as a reducing agent. The treatment of the coupons was then performed as previously described [25,26]. Before treatment, the coupons were sterilized by spraying them with ethanol 70% (wt/wt in deionized water) and exposure to UV radiation (20 minutes on each face). The samples were then placed into 50 mL serum bottles, and autoclaving was performed under anoxic conditions (as defined in pre-incubation). Next, 20 mL of bacterial solution or culture medium (abiotic control) was added. After 7 days of incubation, the coupons were taken out of the treatment solution and sterilized as above (no more bacteria or culture media were present on the treated surfaces). All of the experiments were performed in triplicates, and the results presented here were identical for each set of samples. 

### 2.3. Analytical Techniques

After treatment, the coupons were sampled and cold-embedded in methacrylate resin using the EpoFix Kit (Struers GmbH—Zweigniederlassung Schweiz, Birmensdorf, Switzerland). Cross-polishing was performed using successive silicon carbide abrasive papers (250, 500, and 1000 grit) and Micro-Mesh abrasive cloths (1800, 2400, 3200, 3600, 4000, 6000, 8000, and 12,000 grades). The cross-sectioned samples were then analyzed with optical and scanning electron microscopy, as well as with Raman spectroscopy.

#### 2.3.1. Optical Microscopy

Microscopic observations were performed under a Polyvar MET optical microscope (Leica Microsystems (Schweiz) AG, <br>Verkaufsgesellschaft, Heerbrugg, Switzerland) to characterize the corrosion layer and the biogenic crystals formed. An estimation of the conversion percentage of the original corrosion layer into biogenic crystals was extrapolated with Axio Vision LE^®^ software (version 4.8.2.0, Carl Zeiss MicroImaging GmbH, Iéna, Germany). 

#### 2.3.2. Scanning Electron Microscopy

Scanning Electron Microscopy coupled with Energy Dispersive Spectroscopy (SEM–EDS) mapping was carried out to evaluate the elemental composition as well as the distribution of the corrosion products and biogenic minerals. Coupons were mounted onto stubs using carbon conductive tape to ensure electrical conductivity and were directly analyzed using a environmental scanning electron microscope Philips XL30 ESEM FEG (Thermo Fisher Scientific, Hillsboro, Oregon, USA) equipped with an energy-dispersive X-ray analyzer. Backscattered electron images were acquired at an acceleration potential of 10 to 25 keV and a working distance of 10 mm. For elemental mapping, a resolution of 64 × 50 pixels and a dwell of 1000 were employed.

#### 2.3.3. Raman Spectroscopy

Raman spectroscopy was also performed to study the molecular composition of the corrosion layer before and after bacterial treatment. The analyses were carried out with a Horiba-Jobin Yvon Labram Aramis microscope equipped with an Nd:YAG (neodymium-doped yttrium aluminum garnet; Nd:Y_3_Al_5_O_12_) laser of 532 nm at a power lower than 1 mW. Single-point measurements were carried out with the following conditions: spectral range 100–1600 cm^−1^, 400 µm hole, 200 µm slit, and 10 accumulations of 10 s. Raman mapping was performed on selected areas with the same conditions and a step size of 2.5 µm in the x and y directions. Spectrum correction (automatic baseline correction) and chemical maps were elaborated using LabSpec Raman spectroscopy software suite (version 6, HORIBA France SAS, Villeneuve d’Ascq, France). The identification of the compounds present was based on literature records and a reference spectra library compiled by the authors.

## 3. Results and Discussion

### 3.1. Structure, Thickness and Continuity of the Corrosion Layer

Microscopic observations of the untreated samples revealed a corrosion layer with brown, red and orange-colored compounds (Figure 2a). The corrosion layer of abiotic control coupons had a comparable thickness and color to the untreated coupons (Figure 2b,c). On the outer part of this layer, some blue spots were also detected. However, these did not form a continuous layer (Figure 2c). The formation of these blue compounds on the abiotic control coupons was probably due to an interaction between the iron corrosion products and the buffer solution containing phosphates and carbonates present in the culture medium. On the contrary, after bacterial treatment, the surface color of the iron coupons drastically changed. Indeed, the original corrosion layer disappeared almost completely, and a continuous layer of blue biogenic crystals was observed instead (Figure 2d).

Even if the thickness of a layer of naturally formed corrosion products is uneven, an overall decrease in the corrosion thickness was observed in the coupons treated with bacteria (Figure 3a). In fact, the mean value of the thickness of the corrosion layer decreased from nearly 28 µm (untreated coupons) to about 7 µm on the treated coupons (Figure 3a). The corrosion thickness decrease is due to the dissolutive microbial reduction of the iron phases composing the original corrosion layer. As a result, part of the iron oxyhydroxides is converted into iron biogenic minerals. The results demonstrated that this specific microbial process did convert a part of the original corrosion layer into reduced iron compounds, as least within the 7-day treatment duration. In fact, it is worth mentioning that treatment duration is a key element to assess and that the metal substrate could eventually become corroded if the growing conditions are not carefully set.

Another important feature observed was the continuity of the newly formed biogenic layer. In fact, in order to inhibit corrosion, this layer has to completely cover the remaining original corrosion layer, avoiding further contact of the metal core with atmospheric oxygen and moisture [3]. Microscopic observations of the cross-sections confirmed that after 7 days of incubation, about 92% of the analyzed surface was covered by biogenic crystals (iron phosphates), while only 55% of the original corrosion layer was covered by blue spots on the abiotic control coupons (Figure 3b). Thus, it can be concluded that bacteria are needed to produce a uniform and continuous layer of biogenic minerals and that an abiotic reduction is not enough to achieve comparable results. 

### 3.2. Elemental Composition of the Corrosion Layer 

Elemental mapping revealed that the corrosion layer of untreated coupons was mainly composed of Fe and O (Figure 4a). The same layer (numbered 1) was observed on the abiotic control coupons (Figure 4b). However, an upper layer (numbered 2) mainly composed of S was also detected. In addition, an outermost layer (numbered 3) composed of Fe, O, and P, was detected and corresponded to the areas with blue spots (Figure 4b). Regarding the treated coupons, the same stratigraphy as for the abiotic control was observed, with layer 2, composed of S, superimposed by layer 3, which is rich in Fe, O and P, corresponding to the area covered by the biogenic crystals (Figure 4c). In this case, no discontinuity within layer 3 was observed (Figure 4c). The presence of a sulfur-rich layer under the biogenic crystals is an interesting observation. In fact, a previous study showed the formation of elemental sulfur (S_8_) and partially oxidized mackinawite (Fe^2+^/Fe^3+^S) on the surface of coupons used as an abiotic control, but not when incubated with bacteria [26]. Analyzing the cross-sectioned samples, the current study demonstrates that a layer mainly composed of S is also present on the bacterially treated coupons, and is localized between the remaining original corrosion layer and the biogenic minerals. This sulfur-rich layer is probably the result of an abiotic reaction between Na_2_S added to ensure anoxic conditions and the reactive corroded surface of the iron coupons. Since this layer is located underneath the biogenic crystals, it can be assumed that it was produced first. It is worth mentioning that the formation of such a layer is already reported during iron corrosion in anoxic environments [32]. Even if the effect of sulfur on the corrosion process of iron is still under evaluation, experimental evidence suggests that elemental sulfur could speed up the corrosion rate of iron objects [33,34]. Indeed, this compound is known to be highly reactive, oxidizing organic and inorganic material regardless of the oxygen content [34]. In the presence of humidity, iron and steel surfaces exposed to elemental sulfur are corroded by an electrochemical reaction involving the reduction of elemental sulfur coupled with the oxidation of iron [33,34]. Hence, the sulfur-rich layer detected here and localized underneath the biogenic layer, if elemental sulfur, could be detrimental for the objects, and has to be avoided. In conclusion, for future improvements, Na_2_S should be replaced by other reducing agents containing less sulfur, such as cysteine [35] or titanium(III) citrate [36].

As previously reported, the formation of biogenic iron minerals containing phosphorus can be attributed to the microbial reduction of iron phases, and the reaction of Fe^2+^ ions with phosphates (PO_4_^3−^) added in the buffered bacterial medium [26].

### 3.3. Molecular Composition of the Corrosion Layer 

Single points, as well as areas, were analyzed with Raman spectroscopy, allowing for the identification of lepidocrocite and goethite as the main compounds in the corrosion layer of untreated coupons (Figure 5). 

On the abiotic control, lepidocrocite and goethite were detected, but also siderite and vivianite (Figure 6). Siderite was probably produced as a consequence of the interaction between iron and the carbonaceous sources present in the culture medium (lactate, fumarate, or carbonated buffer) [35]. The formation of vivianite (an Fe^2+^ phosphate) on the abiotic control coupons can be the result of the interaction of the phosphorus buffer with Fe^2+^ ions. The latter can either be present in the original corrosion layer or be produced when the Fe^3+^ phases of the original layer are abiotically reduced by Na_2_S. In fact, the reduction of iron in the abiotic control has already been documented in similar conditions [14]. 

In the samples treated with bacteria, lepidocrocite and goethite were not detected. In fact, after treatment, the thickness of the original corrosion layer drastically decreased, making its detection difficult by Raman spectroscopy (spatial resolution of about 1 µm). As observed above during SEM–EDS analyses, these iron corrosion compounds have been converted into biogenic crystals through microbial reduction. These newly formed minerals were identified as a mixture of two different iron phosphates. The most abundant was vivianite (Figure 7). The vivianite spectrum displayed an intense vibrational band at 949 cm^−1^ and two less intense vibrational bands at 1014 and 1052 cm^−1^, typical of the P-O stretching mode [37]. In order to localize this compound, its characteristic Raman shift at 949 cm^−1^ was employed for chemical mapping (Figure 7). Interestingly, another iron phosphate compound identified as barbosalite Fe^2+^Fe^3+^_2_(PO_4_)_2_(OH)_2_ was also detected. The same bands related to the P-O stretching mode were present but with different relative intensities. For barbosalite, the main Raman shift was at 1015 cm^−1^ (Figure 7). Since barbosalite contains both Fe^2+^ and Fe^3+^ ions, its formation could be the consequence of the interaction between Fe^3+^ ions present in the original corrosion layer, Fe^2+^ ions already present or produced from microbial iron reduction, and the PO_4_^3−^ ions contained in the buffer solution. Finally, Raman measurements did not allow the detection of the sulfur-rich inner layer revealed by SEM–EDS. This could be explained by the low thickness of this layer, probably below the spatial resolution limit of Raman spectroscopy (about 1 µm).

## 4. Conclusions

The development of new and ecologically friendly strategies to protect outdoor iron surfaces has a clear economical, as well as ecological, interest. We demonstrated the potential of bacteria as an alternative for the development of innovative and green methods to protect iron artifacts from detrimental corrosion [24,25,26,27,28]. Here, the obtained results allow us to better understand how reactive corrosion layers are converted into biogenic iron phosphates with *Desulfitobacterium hafniense*. These biogenic minerals covered almost all of the remaining original corrosion layer. This layer could act as a barrier isolating the unstable iron corrosion products that would eventually still be present underneath from the exposure to atmospheric oxygen and moisture that could lead to further corrosion. Vivianite is in fact a stable, poorly soluble, and non-oxidizing Fe^2+^ mineral [22]. However, great attention has to be drawn to the composition of the bacterial solution applied, which not only drives the type of biogenic minerals produced, but also potentially contaminates the corrosion layer with undesired compounds that are able to instigate further corrosion, such as sulfur-containing compounds. In fact, through the stratigraphy study carried out here, a sulfur-rich layer was detected below the biogenic iron phosphates. This layer was not detected with surface analyses of the coupons, and only stratigraphic investigations allowed us to conclude that careful attention has to be paid to the culture medium composition in order to produce a stable vivianite layer that would passivate the iron surface. Our study on cross-sectioned samples further improved the evaluation of the depth efficiency of the proposed bacterial treatment, as well as demonstrated the formation of biogenic vivianite as an adherent, even, and uniform layer. These features are important criteria when designing new protective systems to provide long-term inhibition of corrosion on iron surfaces.

## Figures and Tables

**Figure 1 materials-13-01176-f001:**
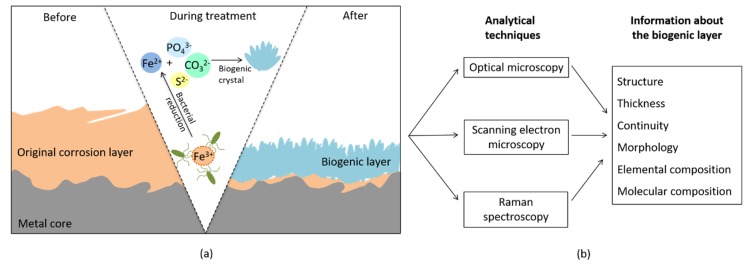
(**a**) Schematic cross-section of a corroded iron coupon submitted to bacterial treatment, showing microbial-induced modifications occurring; (**b**) the analytical methodology performed.

**Figure 2 materials-13-01176-f002:**
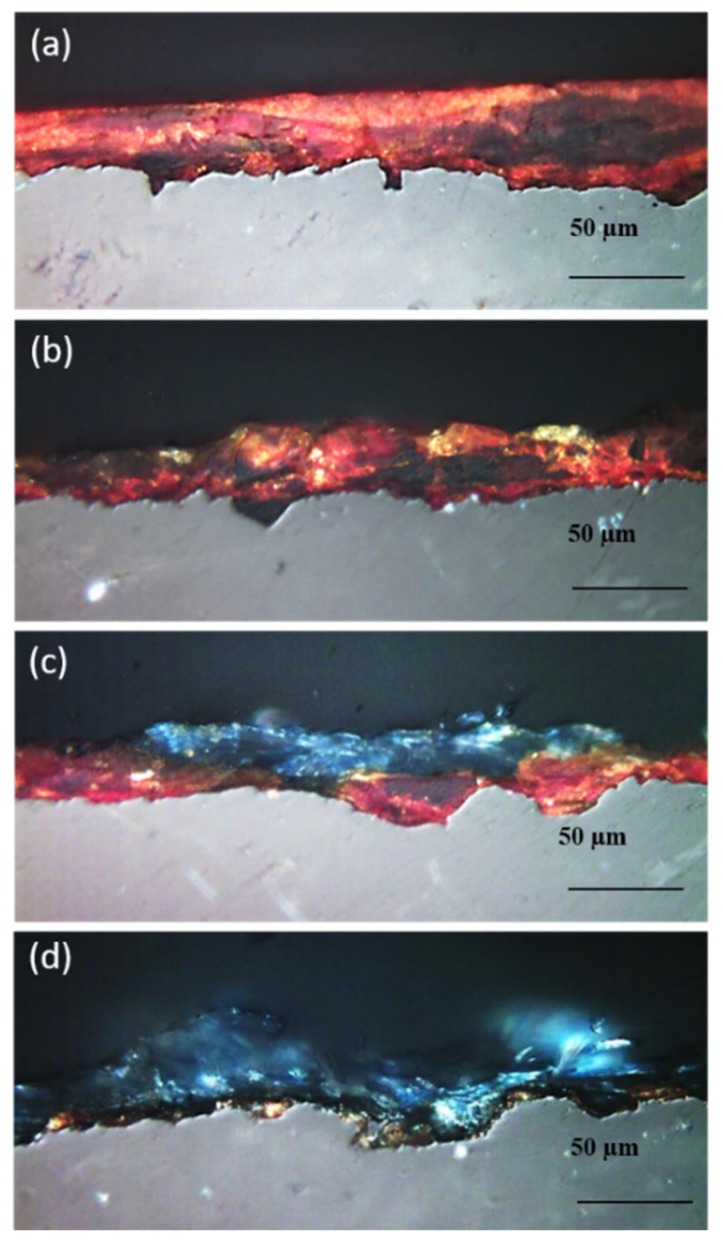
Optical microscope images of untreated (**a**), two different zones of the abiotic control (**b**,**c**), and bacterially treated (**d**) iron coupons.

**Figure 3 materials-13-01176-f003:**
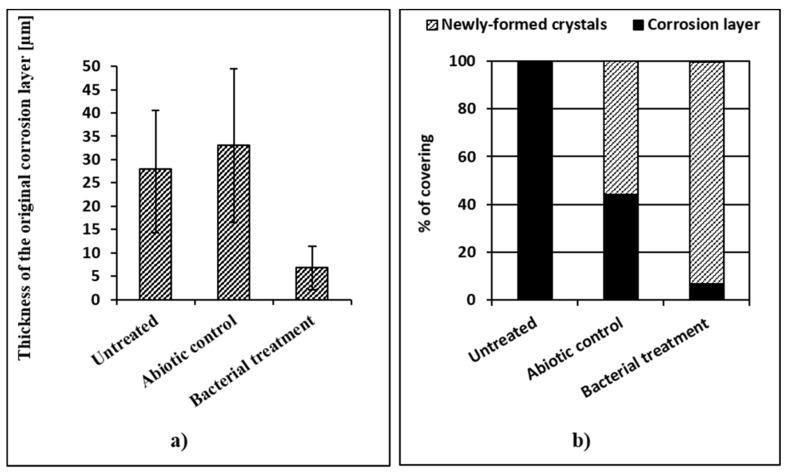
Efficiency of the treatment in terms of thickness and surface covering: (**a**) a graphic representation of the thickness of the corrosion layer of untreated, abiotic control, and bacterially treated iron coupons; (**b**) an estimation of the surface covered by iron phosphates by microscopic observation of the cross-sections.

**Figure 4 materials-13-01176-f004:**
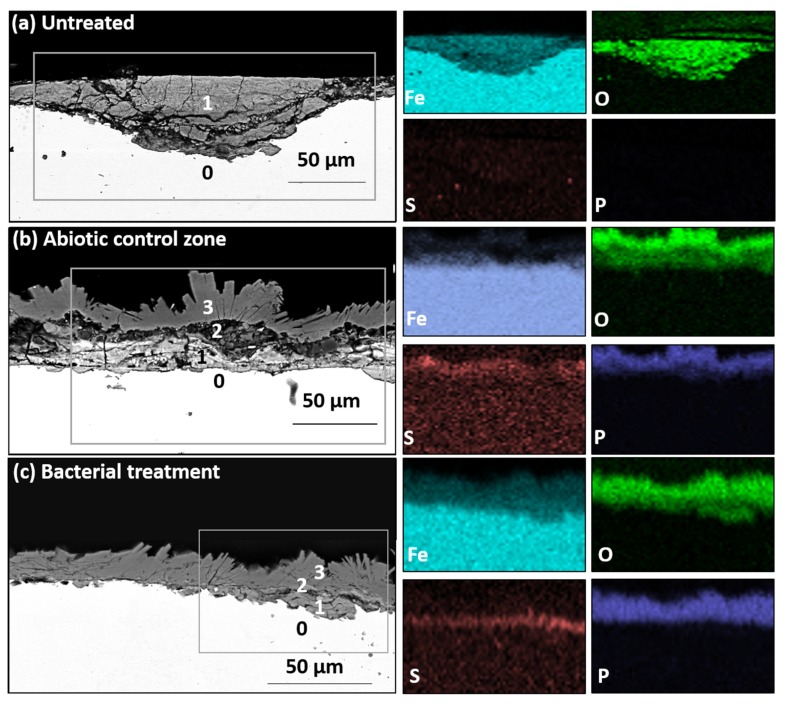
SEM–EDS characterization of (**a**) untreated, (**b**) abiotic control and (**c**) bacterially treated iron coupons. On the left, images of backscattered electrons with a square box indicating the analyzed area and numbers indicating the different layers present: 0 = bulk metal; 1 = original corrosion layer; 2 = newly formed S-rich layer; 3 = biogenic layer. On the right, elemental mapping showing the presence of iron, oxygen, sulfur, and phosphorus in the different layers.

**Figure 5 materials-13-01176-f005:**
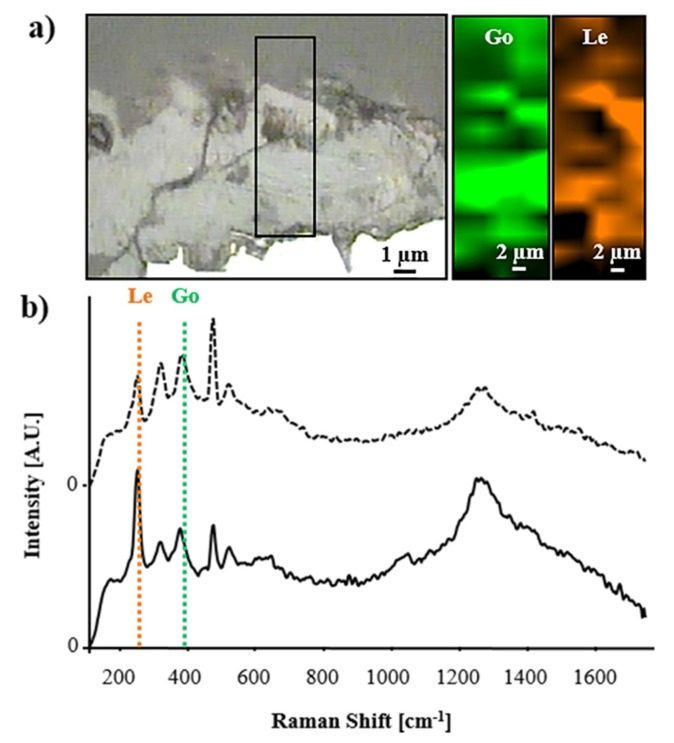
Molecular mapping performed by Raman spectroscopy on untreated coupons. (**a**) From left to right, a microscopic image with a square box indicating the analyzed area and chemical maps of goethite (Go) and lepidocrocite (Le). (**b**) Representative Raman spectrum of a mixture of lepidocrocite and goethite with the corresponding spectral regions selected for the elaboration of the respective chemical maps of lepidocrocite (region labeled “Le” with orange dashed lines) and goethite (region labeled “Go” with green dashed lines).

**Figure 6 materials-13-01176-f006:**
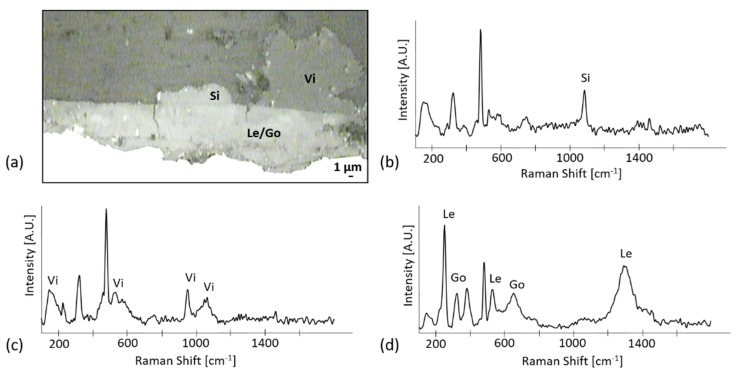
Raman characterization of abiotic control coupons: (**a**) Microscopic image of the analyzed area with identified compounds indicated as siderite (Si), vivianite (Vi), and a mixture of goethite (Go) and lepidocrocite (Le). Representative Raman spectra of the different analyzed regions where (**b**) siderite (Si), (**c**) vivianite (Vi) and (**d**) a mix of goethite and lepidocrocite were identified (Le/Go), respectively.

**Figure 7 materials-13-01176-f007:**
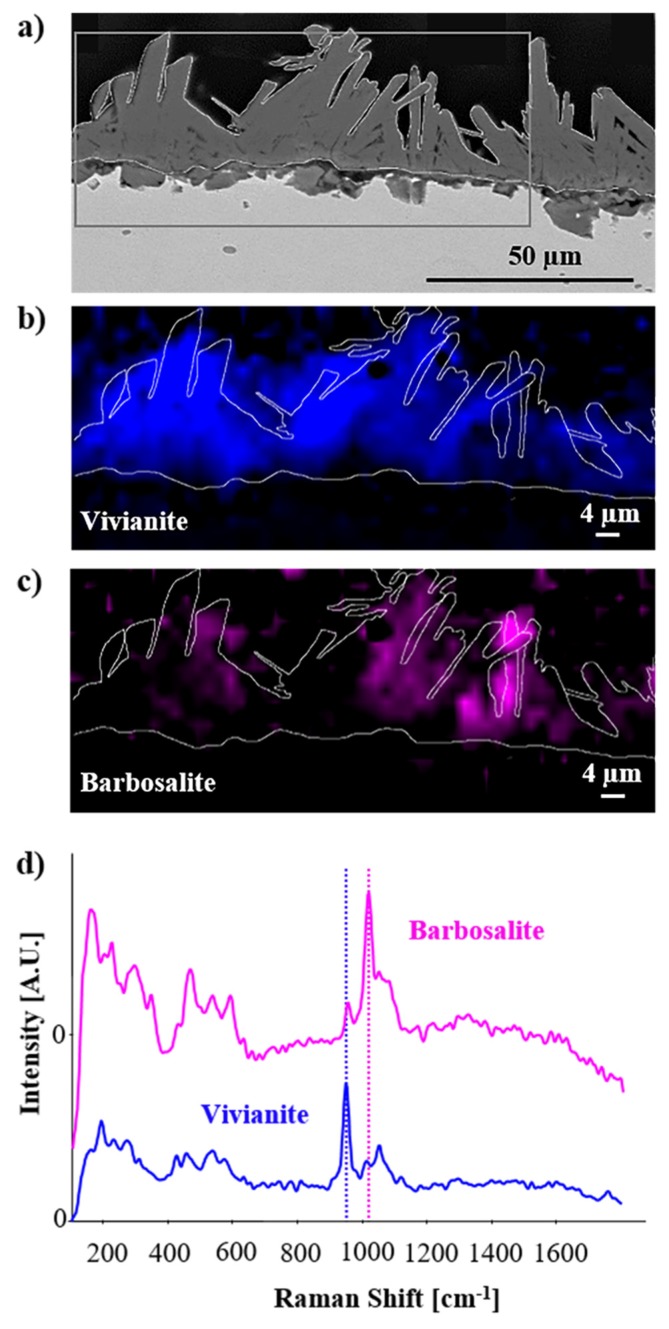
Molecular mapping performed by Raman spectroscopy on bacterially treated iron coupons: (**a**) image of secondary electrons with a square box indicating the area analyzed by Raman spectoscopy, chemical maps (**b**) of vivianite (Vi, blue) and (**c**) of barbosalite (Ba, pink), and (**d**) Raman spectra of vivianite and barbosalite with the corresponding spectral regions selected for the elaboration of the respective chemical maps of vivianite (indicated with blue dashed lines) and barbosalite (indicated with magenta dashed lines).

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
