# Peer review of "Investigation of Biogenic Passivating Layers on Corroded Iron"

_materials, 2020, doi:10.3390/ma13051176_

Round 1

Reviewer 1 Report

The manuscript deals with surface treatment of corroded steel by specific bacteria, which were shown to convert the original layer of corrosion products into a phosphate-based mineral. The paper is well written and contains all necessary elements including comprehensive summary, detail experimental part, clear description of important observations and correct conclusions. The topic is interesting and although treatment of objects of cultural heritage by the described technique is probably quite far, the paper will be helpful for other research teams active in the same area. I recommend it for publication in Materials. Further minor remarks follow:

  • Line 44. How does chloride mechanically damage objects? Chloride-based compounds can disintegrate layers of corrosion products, but an object itself? Please, explain.
  • Line 65. Vivianite and other minerals are mentioned several times before their formula is disclosed (line 240 for vivianite). It should be done when mentioned for the first time. It might be even useful for readers to give a table with chemical composition and eventually other information on the minerals of interest.
  • Line 84, abstract. “Presenting” is not the right expression here. Simple “with”, or “covered with”, would do better.
  • Line 96. Reducing instead of reductive might be more correct.
  • Line 149 and further. “The corrosion thickness” is not the right term here. It is the thickness of the layer of corrosion products, or similar.
  • Line 154-156. “In opposition to the general opinion regarding MIC, the results demonstrated here that this specific microbial process did not induce corrosion by attacking the metal substrate…”. This is not really proved in the paper. It is demonstrated that the bacteria converted most part of corrosion products to phosphate-based minerals. However, there is no proof that the bacteria did not corrode some of the steel substrate too. It would be necessary e.g. to measure mass loss before and after the treatment to show this.
  • Line 157. I do not know what “fironact” means.
  • Line 172. Figure 3b shows a ratio of iron phosphate. However, it is not disclosed above that the blue layer is composed of iron phosphate. It would be more proper to call it “converted layer” or similarly here.
  • Line 265. The fact that iron phosphates are more stable is not proved in the paper. The statement is based on previous works. It does not belong to conclusions.
  • Line 277-278. It is not demonstrated that the biogenic layer of vivianite is adherent. No such data are given in the paper.
  • Line 287. Remove “please add”.

Author Response

We would like to thank you for giving us the opportunity to respond to the reports of the referees. Below we provide responses to their comments (our responses in red and italic) and indicate the changes we have made in the manuscript (in blue).

Reviewer 2 Report

-

Author Response

We would like to thank you for giving us the opportunity to respond to the reports of the referees. Below we provide responses to their comments (our responses in red and italic) and indicate the changes we have made in the manuscript (in blue).

n/a

Reviewer 3 Report

General comments and suggestions

The interesting investigation looks at the mechanism of formation of biogenic minerals on corroded iron surface due to the actions of bacterial activities. Desulfitobacterium hafniense strain TCE1 is used for this particular purpose on corroded iron samples under urban atmospheric conditions. Surface and stratigraphic analyses on cross-section samples were carried out using a combination of microscopic and spectroscopic methods. The severe corrosion of iron-based materials upon exposure to diverse conditions, both in the general-purpose use and archaeological\cultural heritage contexts, and their conservation treatments are very challenging tasks. Effective interventions with long term protection capabilities that take into account respecting the integrity of the iron materials, and at the same time, the environmental impact of the treatments, are rare. There are some attempts to use biogenic materials for protection of non-corroded iron objects, but not for those that have already undergone corrosion. Even though actions of certain species of bacteria on iron materials can be harmful, exacerbating the corrosion advance, other species are indicated to offer protection via different mechanisms, including iron reduction and biogenic mineral formations as described in this contribution. The systematic examination of the effects of the later types, on already corrode materials to study the mechanisms involved, is an important contribution of this work reported. It is based on previous experiences of the investigators on the study of different bacterial species with respect to their capability to produce biogenic minerals and potentially consolidate already corroded outdoor and archaeological materials. Especially, the cross-sectional examinations of the samples treated by the microbes, in addition to the surface analysis reported earlier, are interesting approaches that provided more information about the mechanisms involved and products formed.

Elemental analysis, using EDS, and molecular ones, based on Raman-spectroscopy, are carried out. Further characterization of the stratigraphy, such as more molecular mapping, and investigation of the mineralogical composition of the surface products introduced (XRD, for example), can shed more light about the different chemical species formed. In addition to the biogenetic materials produced, mainly due to the actions of the microbes employed, the effects of the different chemical used in the culture medium for the purpose of buffering the reaction solution, creating the reducing environment and passivating condition could to be systematically studied. They seem to have profound effects. What would be the nature of the mineral formations and the mechanisms of consolidation if other materials were used? The sulphur-rich layer reported could be damaging to the future stability of the treated iron objects. Ways need to be devised to prevent it through closer examination of the origin, cause and mechanism of its formation. If only the elemental analysis is the basis of the characterization of the product associated with elevated levels of sulphur, in what form is it existing? All elemental sulphur? some sulphide(s)? or could there be other chemical species? Are there carbonate formations detected? The study of formation of the sulphur-containing layer and its characterization (composition and structure) would be interesting for a better insight. In fact, the same holds true to the one labelled as biogenic layer that appears to be heterogenous in composition and structure. Could there be other species not detected besides those reported?

How this treatment would behave upon exposure to ambient conditions? It could be interesting to evaluate how treated materials change or remain intact when exposed to ambient conditions following their treatment. Probably, it can be a focus of future investigation. Test on larger samples could also possibly be carried out to allow characterization of the different components formed in the various layers of the stratigraphy of the cross-section, using diverse methods including XRD. The effects of time on evolution of the transformation of the interface between the culture medium and the corroded iron could also be studied for a better insight. Would there be more and more formation of the blue iron phosphates- and black sulphur-rich layers with time (for example)?

Specific comments and suggestions

Line 55. Can conservation treatments related to archaeological materials be covered in this section briefly highlighting their pros and cons? Archaeological objects are considered in this work as one of the potential areas of the tested treatment.

Line 80. The well-planned methodology in the experiment is clearly and nicely illustrated along with the mechanisms involved in the bacterial action on the corroded iron surface. Where is the sulphur-rich layer and the other layers? As illustrated by the microscopic images, there are well-defined layers in the one generally classified as biogenic layer. How about the atmosphere under which the experiment was conducted?

Line 85. The plate was exposed for about ….

Line 88. What was the basis for selecting D. hafniense strain TCE1 for this study as other bacterial species were also found to lead to formation of similar bio-minerals on iron samples corroded under urban conditions?

Line 96. The reducing agent (Na2S) could be one of the sources of the sulphur-rich layer noted in the result section, that could have a negative impact on the long-term consolidation effect of the bacterial treatment. Could there be an attempt to use another agent with similar redox characteristics?

Line 103. Triplicates were used. Were the results identical from each set?

Line 121. Were the measurements carried out in both SE and BSE modes? Some of the images show contrasts in the elemental compositions, implying BSE imaging. How about the vacuum condition? Since no coating was used to ensure electrical conductivity of the samples, environmental mode (not high vacuum) might have been used. If so, that could be specified in the experimental section for the sake of clarity.

Line 131. Which reference spectra library was used? In-house compiled? From a supplier?

Line 141. Basis for this implication? Sulphur-based compounds indicated from analysis of this layer? What looks like the heterogeneity in the chemical composition of this layer? Is it made up of one type of chemical species?

Line 147. Is the blue layer in 2c same as that in 2d? If so, could the reductant has a greater role here compared to the biogenic effect from the bacterial treatment?

Line 147. The black layer rich in sulphur? Could it be mostly derived from the reductant used? Could the bacteria be active in fixing it in close proximity to the metal considering the mobility of other ions and formation of the biogenic compounds (iron phosphates, for example)?

Line 154. How much of the iron reduced in the biogenic process is converted to bio-minerals and how much do exist in solution form? Could ion analysis of the solution before and after treatment provide some information? Certain explanation about speciation of the iron could also be acquired in this manner.

Line 157 … at least   within the 7-days….?

Line 157 In fironact?

Line 157. The effect of time could also be studied using more triplicates of samples followed by similar stratigraphic and surface analyses, as well as thorough characterizations of the products formed due to the interaction with the corrosion layer.

Line 163. Some examples of micrographs of these comparisons would be interesting if they can be presented.

Line 165. What are these blue spots\layers, in the abiotic control case, composed of again? Iron phosphates, exactly like the those formed in the bacterial treatment case, or are they different in composition? If so, what is the very special role of the bacteria if the phosphates of iron can also be formed in the absence of them (with the exception of the coverage extent described)?

Figure 170. Why is the thickness of the original corrosion layer in the untreated coupons smaller than that placed under abiotic control? Is thickness measurement an average from the triplicate sample measurements? How are the variations within one cross-section sample taken into account? Similar information acquired from the SEM-EDS analyses? I think more detailed information at better resolutions can be acquired from the later.

Line 197. What is the composition of this sulphur-rich layer in this experiment? Elemental sulphur as reported in the references or could it also exit in different form(s)? Could bacterial action have some role in the deposition of this sulphur-rich layer? Since the reduction of iron by the microbes, and the subsequent mobility of the reduced iron from the corroded layer, is attributed to the formation of the upper-laying biogenic minerals composed of iron phosphates, could this phenomenon be ruled out?

Line 206. Secondary electron images? Phase contrast (elemental composition-based) appears to be discernable. Are they back scatter electron (BSE) images shown to the left? Would carbon mapping be informative if included (like carbonates, for instance, since siderite was mentioned as one of the potential bio-mineral formation from the previous studies)? What were the elements detected in the EDS spectra collected from the sulphur rich layer? Do they all imply elemental sulphur? Was there any indication of the formation of siderite from the Raman investigation?

Line 214. Shifts in labelling of the peaks in the Raman spectra? They do not show the peaks of goethite and lepidocrocite as intended. Probably needs redrawing.

Line 223. It would be great if the effects of the diverse components in the culture media are systematically studied in the future along with the effects of the bacterial treatment, since there are indications of the formation of the sulphur-rich and iron phosphate-rich layers even before the introduction of the selected bacteria.

Line 250. What was the elemental profile in the EDS spectra collected from this layer? Could that be a good indicator of elemental sulphur as suspected or others (some compounds of sulphur)?

Line 253. Again shifts observed in the Raman spectra labelling.

Line 265. How would this treatment behave upon exposure to ambient conditions? This could be interesting to evaluate how treated materials change under normal conditions following their treatment. Could probably be a focus of future investigation.

Line 267. Inhibition of exposure to atmospheric oxidation and moisture: Was this specifically tested and examined?

Line 271. This is an important observation that could be addressed in future experimental designs to explore other replacements. However, even the study of formation of this layer and its characterization (composition and structure) would be interesting for a better insight, as pointed out in the general comment.

Author Response

(The authors gave the same response as above.)
